# Change of Residents' Attitudes and Behaviors toward Urban Green Space Pre- and Post- COVID-19 Pandemic

Luyang Chen [1], Lingbo Liu [1], Hao Wu [2], Zhenghong Peng [2,*] and Zhihao Sun [1]

1  Department of Urban Planning, School of Urban Design, Wuhan University, Wuhan 430072, China; chenluyang@whu.edu.cn (L.C.); lingbo.liu@whu.edu.cn (L.L.); sun99xt@whu.edu.cn (Z.S.)
2  Department of Graphics and Digital Technology, School of Urban Design, Wuhan University, Wuhan 430072, China; wh79@whu.edu.cn
*  Correspondence: pengzhenghong@whu.edu.cn; Tel.: +86-27-6877-3062

**Abstract:** The COVID-19 pandemic has changed and influenced people's attitudes and behaviors toward visiting green spaces. This paper aims to explore the association between residents' health and urban green spaces (UGS) through an in-depth study of changes in residents' use of UGS under the influence of the COVID-19 pandemic. The Wuhan East Lake Greenway Park was selected as the location for the field survey and in-depth interviews. At the same time, an online survey was also conducted (total number = 302) regarding participants' physical and mental health and their attitude and behavior toward the UGS. A paired sample *t*-test and binary logistic regression were performed to investigate the association between participants' health and UGS during COVID-19. The results show that: (1) the COVID-19 pandemic has primarily changed the leisure patterns of parks, with potential impacts on the physical and mental health of participants; (2) the purpose, frequency, timing, and preferred areas of participants' park visits have changed to varying degrees after the pandemic, highlighting the important role and benefits of UGSs; (3) the physical and mental health of participants and urban development issues reflected by UGS use are prominent. This study reveals that awareness of the construction and protection of UGSs is an important prerequisite for ensuring the health of urban residents.

**Keywords:** postpandemic era; urban green space; green landscape; residents' health

## 1. Introduction

There is a growing body of literature that recognizes the importance of urban green spaces (UGS) for residents' lives and well-being [1–4]. The rapid outbreak of novel coronavirus (COVID-19) pandemic has fundamentally changed people's lives and greatly impacted their daily lives [5,6]. Although most studies have focused on morbidity and mortality studies related to the disease, the physical and mental health [7,8] aspects of the residents are worthy of more study [9].

UGSs provide multiple ecosystem services [10] and are of great value. Not only do they benefit human physical and mental health and social development, but these roles may be amplified in special times [4,11,12]. A lot of pieces of evidence have shown that the unavailability and absence of UGSs, social isolation, and blockades during COVID-19 have a negative impact on mental health [13], presenting varying degrees of anxiety, anger, fear, irritability, reduced well-being, and other negative emotions [14–16], and increasing the risk of mental health disorders [17]. On the other hand, exposure to blue–green space is beneficial for physical and mental health [18,19], relieves stress [3], reduces anxiety, improves attention recovery [20], and increases well-being and satisfaction. To some extent, it increases motivation to exercise and enhances physical activity, thus reducing diseases such as obesity, hypertension, and hyperlipidemia [21–23]. A large proportion of the population believes that the pandemic greatly affected physical activity levels and physical

health, and tries to restore physical function and relieve psychological stress by increasing walking exercise [24–27]. The aforementioned positive and negative effects are closely associated with the resident and ultimately manifest visually as changes in resident use behaviors and attitudes.

Behaviors and patterns of UGS use changed quietly as a result of the COVID-19 pandemic. During the Tokyo Shinkansen epidemic, older adults, elementary school students, and others experienced lifestyle changes, lack of exercise, stress accumulation, and decreased well-being [15]; park use and associated mental health patterns among college student populations during the pandemic raised concerns, with some studies suggesting that young people were more likely to experience negative emotions such as stress, anxiety, and depression [13,28–30]. Some studies have shown significant increases in park visitation [4,31] and increased park use [6,32,33], while some urban parks have seen decreases in visitor numbers [34], decreases in public space engagement, and reduced risk of epidemic prevention and control with the use of public urban green spaces [35]. These conflicting and changing uses need to be studied by more in-depth investigations.

The pandemic is reflected in differences in attitudes and behaviors of different groups in UGSs [26]. Additionally, the uneven distribution of UGSs at particular times can also cause changes in user behavior. Wuhan was one of the first cities to experience a major outbreak of the COVID-19 pandemic, and it went through a three-month lockdown. To understand the actual changes in the use of UGSs before and after the pandemic, we conducted a questionnaire survey in Wuhan's largest, most environmentally friendly, and most well-known green space, called East Lake, after the announcement of Wuhan's "lifting of the city lockdown" on 18 April 2020. The COVID-19 erupted in January 2020 and Wuhan lifted the lockdown in April 2020. Therefore, in the study, the "pre" refers to the time before January 2020, while the "post" refers to the time after April 2020. This survey intends to examine the changes in the relationship between UGSs and residents' health under the COVID-19 pandemic from the perspective of changes in residents' use behavior. Survey analysis was conducted to answer the above hypotheses based on the specific performance of residents of different ages, genders, occupations, and statuses in terms of their attitudes and behavioral patterns of use. First, sociodemographic information was collected to classify the population of users; then, the specific changes in residents' use of urban parks were used to study the deeper associations during the pandemic; and finally, the changing patterns of attitudes and behaviors of use were explored to consider the construction of UGSs.

## 2. Research Methods and Data Collection

### 2.1. Questionnaire Survey

To investigate the specific changes in residents' use of UGSs before and after the COVID-19 pandemic, a combination of field and online surveys was conducted in the paper. For the field survey, the respondents were selected from the core areas of five different scenic spots in East Lake and filled in voluntarily for 6–10 min by the respondents in an informed manner. During this field survey, a total of 297 questionnaires were distributed. The online method is more effective and easier for people to visualize the various options. An online questionnaire was distributed via WeChat to groups who had visited East Lake to ensure that the groups researched had realistic feelings about the use of the UGS. As a result, a total of 38 participants responded to the online questionnaire. The choice of an online survey is based on three main considerations. (1) Due to the development of the times, the popularity of digital devices and the internet has changed the daily life of a large part of the population, and the internet is increasingly used, especially by young people, to reach a large number of people through social media (WeChat) posting and forwarding. (2) Field surveys are often influenced by many factors, while an online survey is more convenient and faster to disseminate. (3) The no-contact approach is more popular due to epidemics. However, the online survey cannot comprehensively collect information from all kinds of people, especially those who cannot use electronic devices

(elderly, students, etc.), resulting in insufficient data samples. Moreover, this method also cannot conduct interviews. Therefore, we decided to conduct multiple field surveys to obtain more realistic and effective information.

The questionnaire was designed to assess three main types of information: sociodemographic, self-ratings of the residents' physical and mental health, and the use of UGSs. In the first part, sociodemographic information was collected to map the personal situation of the residents to connect with the information that follows [1,36,37]. In the second part, the self-rated questionnaire was used to capture the physical and mental health status of the participants, and then the results were classified and analyzed [36,38]. In the third part, six factors, including purpose of visit, frequency of visit, mode of transportation, mode of travel, area visited, and duration of stay, which is the most intuitive and reflective of changes in UGS use, were selected for question setting to collect changes in use [6,39–42]. Table 1 shows the questionnaire content of residents' self-rating of their physical and mental health, as well as their attitudes and behavior toward green space. For the above self-rated questionnaire, the five-point Likert Scale Method was introduced to capture the intensity of participants' feelings for a given item, which means participants would choose one of five levels (for example, 1 = worse, 2 = poor, 3 = general, 4 = good, 5 = better) for a series of statements on a symmetric agree–disagree scale.

This round of research was conducted from March 2021 to November 2021, and the questionnaires were distributed and collated during the period when the pandemic was relatively stable and the contrast between before and after the COVID-19 pandemic was more obvious. A total of 335 questionnaires were distributed during this period, and 302 valid questionnaires were returned, with an efficiency rate of 90.1%.

*2.2. Field Trips*

2.2.1. Location Selection

The Wuhan East Lake Greenway, which has a superior natural environment and convenient transportation, was chosen as the research object. It is located in the Wuhan East Lake Scenic Area which is situated in the eastern part of the city. It is the largest internal urban park in Wuhan, serving a large part of the city, and is a representative of UGSs with a large volume.

The planning and construction are based on "Eco-Wuhan" to create a world-class greenway around the lake. Relying on humanistic and historical resources, rich in natural resources, Bruno Deacon, an official of UN-Habitat, called it a model. With a total length of 101.98 km and a width of 6 m, the East Lake Greenway connects five scenic spots of the East Lake, which is shown in Figure 1. With its unique natural scenery, historical and cultural heritage, biological diversity, plant diversity, and functional diversity, it attracts more than 40 million people and has become a preferred place for residents to travel, relax, and interact outdoors.

2.2.2. Field Research

Considering the general context of the pandemic and the specificity of public space use, priority was given to on-site behavioral observation, questionnaire distribution, targeted in-depth interviews, green space landscape classification, local staff interviews and consultations, and photo documentation to collect on-site information. In the on-site research, the following two aspects were specifically addressed. First, the participants were classified using the observation method to determine whether they were permanent residents by their behavior. For example, if residents exhibit behaviors such as excessive photo-taking, sightseeing, and punching in, we classified these residents as nonlocals, and they were filtered out in our questionnaire. Second, after the interviewees were identified, inquiries were made and targeted interviews were conducted. Finally, the identities of all the interviewees were classified, and many effective interview results were obtained.

**Table 1.** Measurement of residents' physical and mental health and their attitude and behavior toward green space.

| Part 1: The Use of Green Space. | | | | | | | | | | |
|---|---|---|---|---|---|---|---|---|---|---|
| | | "Pre-Pandemic" | | | | | "Post- Pandemic" | | | | |
| 1 | How do you think your physical health? | Worse | poor | general | good | better | Worse | poor | general | good | better |
| 2 | How do you think your mental health? | Worse | poor | general | good | better | Worse | poor | general | good | better |
| 3 | What is the purpose of your visit to the park? | exercise | relax | Relieve the pressure | view | entertainment | exercise | relax | Relieve the pressure | view | entertainment |
| 4 | Have you been here a lot? | hardly | occasionally | sometimes | often | almost everyday | hardly | occasionally | sometimes | often | almost everyday |
| 5 | What is your mode of transportation to the park? | public transportation | private car | walking | cycling | other | public transportation | private car | walking | cycling | other |
| 6 | How do you choose to travel? | alone | community | | | | alone | community | | | |
| 7 | Which areas of the park do you usually choose? | Baima scenic area | Tingtao scenic area | Luoyan scenic area | Moshan scenic area | Chuidi scenic area | Baima scenic area | Tingtao scenic area | Luoyan scenic area | Moshan scenic area | Chuidi scenic area |
| 8 | How long do you stay? | 20 min | 20–40 min | 1 h | 1–2 h | >2 h | 20 min | 20–40 min | 1 h | 1–2 h | >2 h |

| Part 2: The evaluation of green space. |
|---|
| 1. What is your evaluation of the green space around your community? <br> ☐ very dissatisfied  ☐dissatisfied  ☐general  ☐satisfied  ☐very satisfied |
| 2. What is your evaluation of park green space? <br> ☐ very dissatisfied  ☐dissatisfied  ☐general  ☐satisfied  ☐very satisfied |

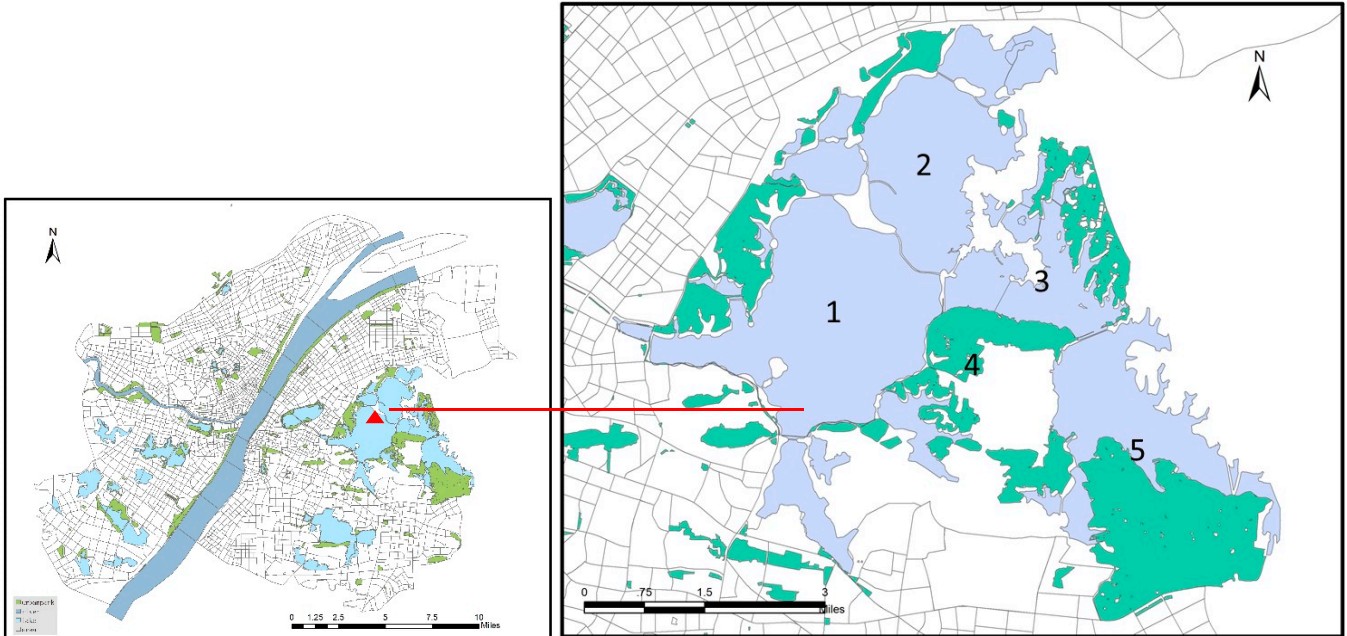

**Figure 1.** Scenic spot distribution map. (1 Tingtao scenic area; 2 Baima scenic area; 3 Luoyan scenic area; 4 Moshan scenic area; 5 Chuidi scenic area).

### 2.3. Statistical Analysis

The descriptive statistics were used to obtain and summarize the sociodemographic characteristics of the participants, their health status, and urban park use before and after the COVID-19 pandemic. For the changes in participants' health and urban park use, the paired sample *t*-test was utilized to quantitatively analyze the impact of the pandemic on them, assuming as a null hypothesis that the pandemic would not affect the participants' health and urban park use.

To analyze the relationship between the existing changes and sociodemographic characteristics of the participants, binary logistic regression was carried out for two trends of changes in residents' health, separately: the increasing trend and the decreasing trend. For the former case, the dependent variable (participants' health increased after the pandemic) was set to 1, and those that remained unchanged or decreased were set to 0. Similarly, for the latter case, the dependent variable (participants' health decreased after the pandemic) was set to 1, and those that remained unchanged or increased were set to 0. Regarding independent variables, several binary variables and categorical scale variables were included. For gender, the male was set to 1 and female was set to 0. For the income trend, the increase was set to 1 and the reduction was set to 0. In other binary variables (including identity, marital, housing ownership, change of income), 1 means yes, and 0 means no. The categorical scale variables (including age, years of residence, education, occupation, income, and evaluation of green space near home or in the park) were used as ordinal value scales.

The paired sample *t*-test and binary logistic regression were performed through the statistical analysis software SPSS 26.0.

## 3. Result and Analysis

### 3.1. Sociodemographic Characteristics

As the central city of central China, Wuhan has 11 million residents [43]. Table 2 shows some basic information about the residents of Wuhan. Among them, males accounted for 50.8%, and females for 49.2%. Excluding the elderly aged over 60, the group aged 30–39 has the highest percentage of the population, at 19.45%. Meanwhile, 33.87% of the residents have bachelor's degrees or above.

**Table 2.** Basic sociodemographic characteristics of Wuhan residents.

| Demographic | Variable | Percentage (%) |
|---|---|---|
| Gender | male | 50.8 |
| | female | 49.2 |
| | <10 | 11 |
| | 10–20 | 7.22 |
| | 20–29 | 11.98 |
| Age | 30–39 | 19.45 |
| | 40–49 | 13.61 |
| | 50–59 | 15.46 |
| | ≥60 | 21.23 |
| | primary Schools | 13.65 |
| | junior High School | 25.34 |
| Education | high School | 19.69 |
| | bachelor's degree or above | 33.87 |
| | other | 7.45 |

source url: The basic sociodemographic characteristics of Wuhan residents were obtained from the Hubei Provincial Statistics Bureau website. (https://tjj.hubei.gov.cn/tjsj/sjkscx/tjnj/gsztj/whs/, accessed on 27 June 2022).

Table 3 shows the sociodemographic characteristics of participants. Of the 335 questionnaires distributed, a total of 302 valid responses were collected with a final response rate of 90.1%. The participation rate of males (62.25%) was much higher than that of females (37.75%), and the respondents were mainly from 20–29 years old, accounting for 41.06% of the total. Most of them are permanent residents (78.15%), and most of them have their own houses (57.95%). Most of the participants are well educated (68.54% with a bachelor's degree or above) and have a stable job. The average monthly income is concentrated in the range of RMB 5000–8000 (30.46%) and over RMB 8000 (30.46%). Due to the impact of the COVID-19 pandemic, the salary of nearly half of the participants (43.71%) has changed, of which 77.27% were less than before the pandemic.

**Table 3.** The statistical results on the sociodemographic characteristics of the participants.

| Demographic | Variable | N | Percentage (%) |
|---|---|---|---|
| Gender | male | 188 | 62.25 |
| | female | 114 | 37.75 |
| | <10 | 0 | 0 |
| | 10–20 | 29 | 9.6 |
| | 20–29 | 124 | 41.06 |
| Age | 30–39 | 70 | 23.18 |
| | 40–49 | 32 | 10.6 |
| | 50–59 | 37 | 12.25 |
| | ≥60 | 10 | 3.31 |
| Identity | permanent residents | 236 | 78.15 |
| | nonpermanent residents | 66 | 21.85 |
| Marital | married | 126 | 41.72 |
| | unmarried | 176 | 58.28 |
| Housing ownership | home ownership | 175 | 57.95 |
| | rental housing | 127 | 42.05 |
| | <1 year | 29 | 9.6 |
| Years of Residence | 1–3 years | 76 | 25.17 |
| | >3 years | 197 | 65.23 |
| | high school and below | 32 | 10.6 |
| Education | technical college | 63 | 20.86 |
| | bachelor's degree or above | 207 | 68.54 |
| | regular occupation | 154 | 50.99 |
| Occupation | freelance | 67 | 22.19 |
| | retired | 13 | 4.3 |
| | current students | 68 | 22.52 |

**Table 3.** *Cont.*

| Demographic | Variable | N | Percentage (%) |
|---|---|---|---|
| | <1550 | 42 | 13.91 |
| | 1550–3500 | 32 | 10.6 |
| Income/month | 3500–5000 | 44 | 14.57 |
| | 5000–8000 | 92 | 30.46 |
| | >8000 | 92 | 30.46 |
| Change of income | yes | 132 | 43.71 |
| | no | 170 | 56.29 |
| Income trend | increase | 30 | 22.73 |
| | reduce | 102 | 77.27 |

### 3.2. Self-Rating of Physical and Mental Health

Numerous studies have confirmed that the presence of UGSs contribute to improved quality of life in many ways. In addition to some environmental and ecological services, cities naturally provide important social and psychological benefits to human society, enriching the meaning and emotion of human life [12]. The visit to UGSs has a positive promotion effect on physical and mental health [44]; conversely, it will also produce a certain degree of fear and pain of negative emotions, and eventually lead to an increase in anxiety, depression, and other mental diseases [45].

In this study, a self-rated questionnaire was used to obtain the physical and mental health status of the participants, and the results are shown in Figure 2. Additionally, the paired sample $t$-test revealed that postpandemic physical and mental health levels have both decreased significantly (for physical health: $t = 5.025$, $p = 0.000$; for mental health: $t = 6.949$, $p = 0.000$). In terms of physical health, the number of participants who felt their physical and mental health status is "fair poor" and "average" has increased, and conversely, the number of participants who felt their health status is "fair good" and "good" has decreased. In addition, the changing trend in mental health also decreased significantly. The result also proved that the pandemic has a certain impact on participants' attitudes and behavior changes in park use and that it is consistent with speculation and existing research that the pandemic has a direct or indirect impact on physical and mental health [18].

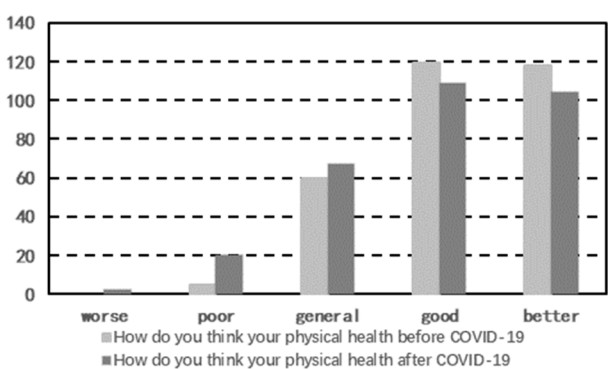 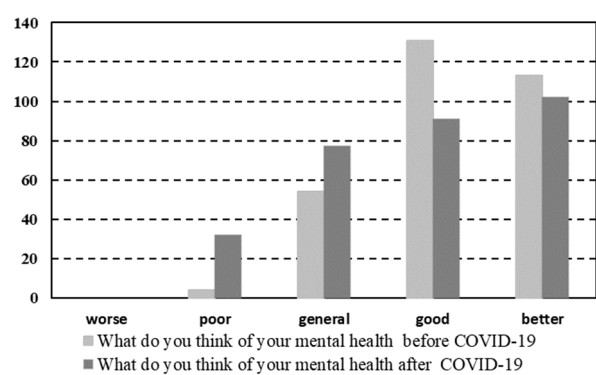

**Figure 2.** Self-rated comparison chart of physical health (**left**) and mental health (**right**).

### 3.3. Urban Park Usage

Changes in the use of urban parks were investigated from four perspectives: visit purpose, visit frequency, accompany mode, and visit duration of the parks, and the results are shown in Figure 3. Additionally, the paired sample $t$-test results of different perspectives are shown in Table 4, in which the visit purpose is not tested because it is a multiple-choice question. These results illustrated that the user behavior and attitude of urban parks changed under the pandemic:

(1) Compared with the changes in the purpose of participants visiting parks before and after the pandemic, the majority of participants choose to "relax", "relieve pressure", and "exercise" after the pandemic, and it is worth noting that "relax" become the first choice. Secondly, the number of participants who choose to "relieve pressure" has increased significantly, which has become an important purpose for participants to go to parks after the pandemic, reflecting the impact of the pandemic on participants' mental health and the enhanced recreational use of UGSs.

(2) In terms of the frequency of visits, it remained stable, but the number of participants who choose to visit "often" and "almost every day" showed a significant increase ($t = -4.421$, $p = 0.000$) in Table 5. Under the impact of the pandemic, the frequency of participants visiting green spaces in parks increased for health reasons, highlighting the important regulatory role of UGSs, especially urban parks, in the context of the pandemic.

(3) The change of visit mode was not significant. It indicated that the number of lone-visitors decreased slightly while the number of visits with multiple companions increased. This phenomenon may be related to the psychological changes of participants after the pandemic.

(4) According to Table 5, The change of visit duration was significant ($t = 7.052$, $p = 0.000$). The number of participants who stay for more than 2 h is significantly more than before the pandemic, which reflects that most residents prefer outdoor activities after the pandemic.

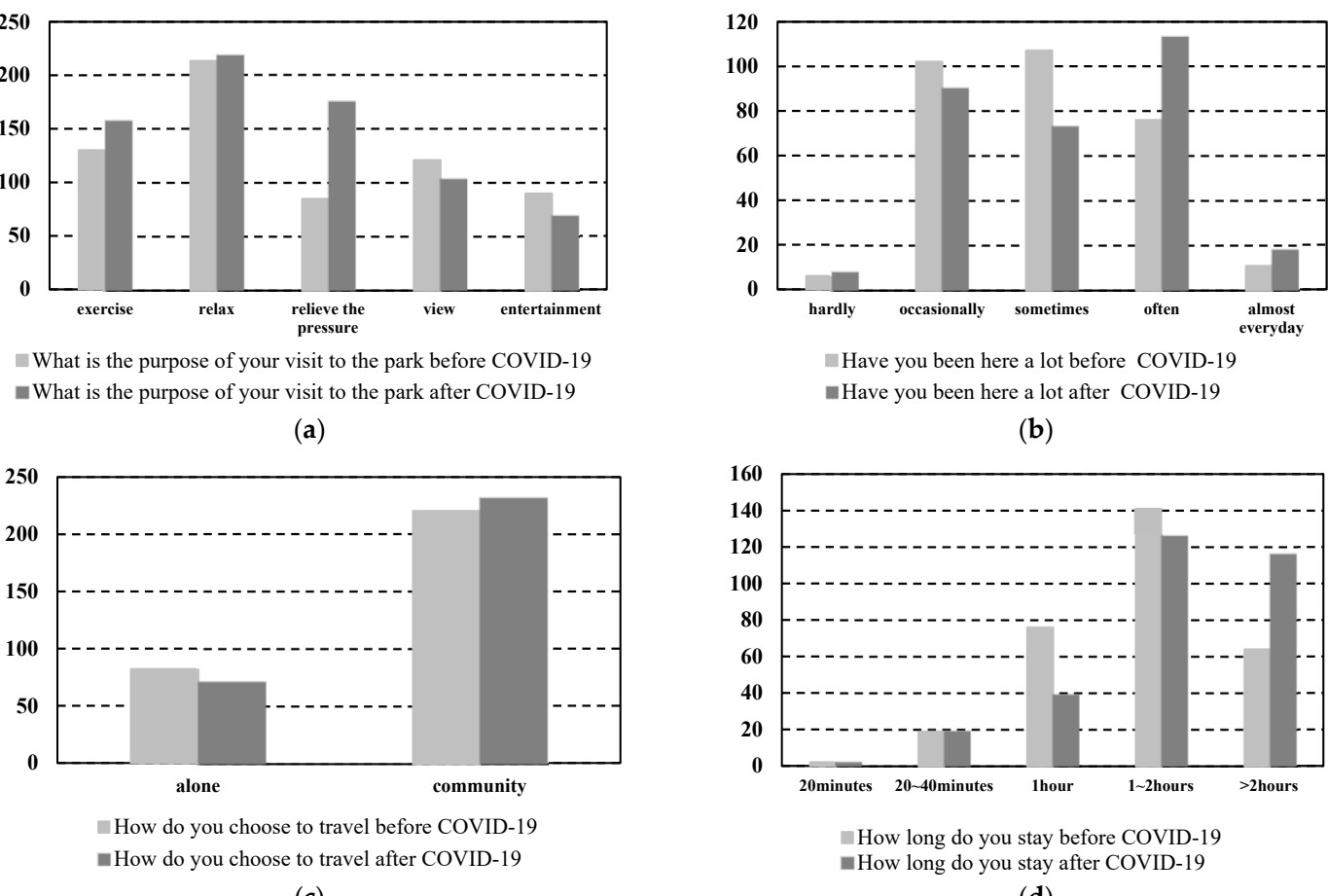

**Figure 3.** Urban park uses during the pandemic: (**a**) purpose, (**b**) frequency, (**c**) accompany mode, (**d**) duration.

**Table 4.** Paired sample *t*-test results of physical and mental health before and after the pandemic.

| | Paired | | Difference (Before vs. Post) | *t* | *p* |
|---|---|---|---|---|---|
| | Before- | Post- | | | |
| Physical health | 4.159 ± 0.795 | 3.970 ± 0.945 | 0.189 ± 0.653 | 5.025 ** | 0.000 ** |
| Mental health | 4.169 ± 0.761 | 3.871 ± 1.002 | 0.298 ± 0.745 | 6.949 ** | 0.000 ** |

$p < 0.05$; ** $p < 0.01$.

**Table 5.** Paired sample *t*-test results of visit frequency, accompany mode, and visit duration before and after the pandemic.

| | Paired | | Difference (Before vs. Post) | *t* | *p* |
|---|---|---|---|---|---|
| | Before- | Post- | | | |
| Visit frequency | 2.947 ± 0.902 | 3.142 ± 1.000 | −0.195 ± 0.768 | −4.421 ** | 0.000 |
| Visit duration | 2.185 ± 0.862 | 1.891 ± 0.903 | 0.295 ± 0.726 | 7.052 ** | 0.000 |

$p < 0.05$; ** $p < 0.01$.

The changes in the intention to visit different areas are shown in Figure 4. According to Figure 4, the number of participants choosing "Moshan Scenic Area" decreased slightly after the pandemic, while the number of participants choosing the other four areas, which are relatively far away, are all increasing. By normalizing the numbers of visits to different areas, it was found that the number of visits to the Moshan scenic area was the highest both before and after the pandemic, while the number of visits to the other four scenic areas was relatively small. Specifically, compared with "before the pandemic", the number of participants choosing the Moshan scenic area decreased "after the pandemic", while the corresponding number of participants choosing the Baima scenic area, Tingtao scenic area, and Luoyan scenic area increased significantly. In terms of geographical location, the Moshan scenic area is located in the core area of the whole park, with convenient transportation and complete infrastructure, and it is a concentrated tourist and leisure area. The Baima scenic spot and the Luoyan scenic spot are located in the northern part of the park with weak geographical advantages. These areas are dominated by ecological wetland resources and have fewer recreational facilities, which may be the main reason for the low number of visits. The Tingtao scenic spot has the largest lake in the East Lake and is dominated by several commemorative scenic spots with low attractions. The Chuidi scenic spot is located in the easternmost part of the park, close to Maanshan Forest Park. It also has rich ecological resources and is relatively secluded. Therefore, the change in visitor numbers before and after the pandemic may be caused by the differences in the location and function of the park. On the one hand, for the consideration of prevention, control, and health, it has gradually become a trend to choose areas with safe distances and safe numbers to avoid potential infection risks. On the other hand, areas with prominent advantages of ecological and natural resources become more attractive to participants.

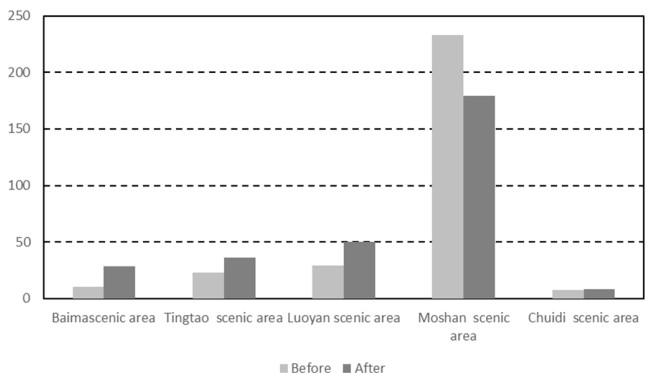

**Figure 4.** Comparison of urban park use areas before and after the pandemic.

*3.4. The Relationship between Changes and Some Independent Variables*

To investigate the influence of factors that are potentially associated with the changes in participants' health or changes in park use status, the binary logistic regression analysis was performed, and the increasing and decreasing trends in these changes were used as dependent variables, respectively, and the social characteristics of the participants were used as independent variables. Results of the binary logistic regression analysis are shown in Table 3 and reveal some distinctive features of the UGS use on participants' health before and after the pandemic. According to Table 6, on the one hand, after the outbreak: the increase in the self-rated physical health of the participants was negatively correlated with age ($p < 0.05$); the increase in the frequency of visiting UGSs was closely related to the quality of green space distribution in urban public spaces (visiting green spaces around parks and neighborhoods) ($p < 0.01$), with both poorer quality green spaces in neighborhoods and better quality green spaces in parks leading to an increase in the frequency of visiting UGS in urban parks; there is a correlation between the change in the mode of escorting more than one person to UGS and participants' income ($p < 0.05$), with higher-income participants preferring to be accompanied by more than one person to UGS; the change in the duration of visiting UGS was significantly correlated with the income level of residents ($p < 0.01$) and showed a negative correlation, and the income level of participants directly led to the change in the duration of visiting UGS.

**Table 6.** The relationship between increasing and decreasing trends and some independent variables.

| | | | B | S.E. | Wald | Sig. | Exp (B) |
|---|---|---|---|---|---|---|---|
| Increase | Physical health | Age | −0.541 * | 0.240 | 5.071 | 0.024 | 0.582 |
| | | Constant | −0.745 | 0.812 | 0.843 | 0.358 | 0.475 |
| | Mental health | | | | | | |
| | Visit frequency | Evaluation of green space near home | −0.843 ** | 0.172 | 24.045 | 0.000 | 0.430 |
| | | Evaluation of green space in the park | 1.077 ** | 0.124 | 19.493 | 0.000 | 2.935 |
| | | Constant | −2.644 ** | 1.030 | 6.585 | 0.000 | 0.071 |
| | Accompany mode | Income | 0.528 * | 0.207 | 6.545 | 0.011 | 1.696 |
| | | Constant | −4.463 ** | 0.893 | 24.997 | 0.000 | 0.012 |
| | Visit duration | Income | −0.442 ** | 0.154 | 8.233 | 0.004 | 0.643 |
| | | Constant | −1.152 * | 0.482 | 5.716 | 0.017 | 0.316 |
| Decrease | Physical health | Gender | −0.658 * | 0.300 | 4.817 | 0.028 | 0.518 |
| | | Evaluation of green space near home | −0.441 ** | 0.158 | 7.759 | 0.005 | 0.643 |
| | | Constant | 1.290 | 0.720 | 3.212 | 0.073 | 3.633 |
| | Mental health | Housing ownership | −0.553 * | 0.272 | 4.139 | 0.042 | 0.575 |
| | | Evaluation of green space near home | −0.568 ** | 0.163 | 12.187 | 0.000 | 0.567 |
| | | Evaluation of green space in the park | 0.708 ** | 0.227 | 9.686 | 0.002 | 2.030 |
| | | Constant | −1.135 | 1.021 | 1.237 | 0.266 | 0.321 |
| | Visit frequency | | | | | | |
| | Accompany mode | Gender | −2.128 * | 1.045 | 4.415 | 0.042 | 0.119 |
| | | Constant | −0.472 | 1.157 | 0.167 | 0.683 | 0.624 |
| | Visit duration | Period of resident | 0.519 * | 0.223 | 5.404 | 0.020 | 1.680 |
| | | Income | 0.183 | 0.099 | 3.408 | 0.065 | 1.201 |
| | | Evaluation of green space near home | −0.330 * | 0.157 | 4.431 | 0.035 | 0.719 |
| | | Evaluation of green space in the park | 0.576 ** | 0.225 | 6.551 | 0.010 | 1.780 |
| | | Constant | −4.120 ** | 1.202 | 11.753 | 0.001 | 0.016 |

* *Sig.* < 0.05; ** *Sig.* < 0.01.

On the other hand, after the pandemic: the self-rated physical health of a part of the population decreased, as there was a negative correlation between the gender of the participants ($p < 0.05$) and the quality of the green space of the living environment ($p < 0.01$); participants who showed a decline in self-rated of mental health were associated with three factors, namely, a negative correlation ($p < 0.05$) between whether they owned their own home and the quality of green space in their residential area, and a positive correlation between the quality of green space in parks and changes in mental health; the shift from

companionship mode to solo mode was correlated with the gender of the participants ($p < 0.05$); the decrease in time spent visiting UGSs was positively correlated with the time spent living and the quality of green spaces in parks, and negatively correlated with the quality of green spaces in the participants' living environment.

## 4. Discussion

The COVID-19 pandemic is still present and will continue to be prevalent. The urban development, physical and mental health of residents, and social changes affected by the pandemic are likely to continue for a longer period of time. It is particularly important to study and discuss the physical and mental health of urban residents. Has the pandemic had an impact on the behavior of urban residents visiting public green spaces? For which specific groups did it have a significant impact? How have they been affected? are the causes of the impact are and the extent of the impact will be the focus of our study. As an urban park consisting of five distinctive urban green areas, East Lake Scenic Area can cover a larger sample of users. Therefore, the results obtained from the survey data we collected are credible and very meaningful. This study focuses on the changes in behavior and attitudes of urban residents using public green spaces during a pandemic to confirm the hypothesized results of the study. Figure 5 illustrates the relationship between UGSs and residents' physical and mental health, as summarized in the study. The first part deals with the use of UGSs, including frequency, purpose, and pattern, which affect the health of the residents through direct or indirect ways, with health manifested as physical, psychological, and social behaviors.

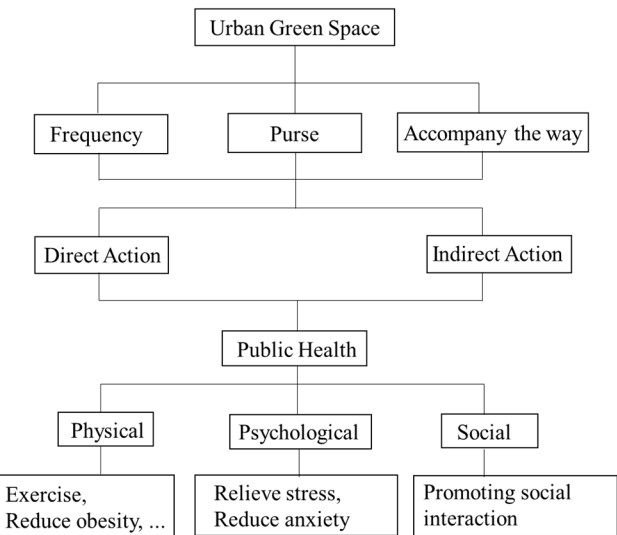

**Figure 5.** The relationship between the factors.

First, the pandemic proved to have an impact on the daily life of the residents. On the one hand, residents' self-ratings of their physical and mental health has changed significantly as a result of the pandemic; on the other hand, specific behaviors in using the garden are significantly different from those before the pandemic. Second, the negative impact of the pandemic on residents' daily life has been confirmed.

(1)   The population visiting UGSs is relatively concentrated. There are more male visitors than female visitors, and the majority of those in the visitor group are young. Among the respondents, the participation rate of men was higher than that of women, and their ages were concentrated between 20–40 years old, indicating that young people are more health-conscious and willing to spend time visiting parks, and confirming that the physical health of some young people has improved after the outbreak. In addition, visitors' behavior was correlated with information about their status and occupation, reflecting that local residents who own their own houses, have long years

of housing, have stable jobs, and are well-educated outperformed other respondents in terms of behavior and health perceptions about visiting green spaces.

(2) The use of UGSs was specific to use behaviors and attitudes. The pre- and post-pandemic dynamics reflected that the pandemic had an impact on the use of park green spaces by urban residents. First, the improvement of participants' health levels after the pandemic was negatively correlated with age; the way of visiting green spaces was correlated with income level; and the change in the duration of visiting green spaces was significantly correlated with income. Second, after the pandemic, the decline in self-rated health status of some people was negatively correlated with gender and the quality of green space in the residential environment; and the decline in participants' self-rated mental health status was negatively correlated with home ownership and the quality of green space in the residential area, while the decline in the self-rated mental health was positively correlated with changes in mental health. Third, it is worth noting that the change in the choice of travel companion mode has a great relationship with gender.

The number of residents choosing to use park green spaces for physical exercise, stress relief, and relaxation increased, which is consistent with existing studies. The predominant focus of choice on physical exercise and relaxation [8] may be related to the negative impact of the pandemic on physical and mental health, which is well-explained by the desire to improve by visiting public green spaces. In terms of frequency of use and duration of use by residents, it is intuitive from the research data that the frequency of use, in general, shows an upward trend, and there is even the phenomenon that some people go almost every day, which may be affected by the quality of urban public spaces and residential green spaces. In terms of the duration of the visits to parks, it is evident that the overall duration of residents is increasing, and there is also a negative correlation between changes in this segment of the population and the income level of residents; there is also a decrease in duration of visits, and the study confirms that it is related to the period of residence, the quality of green spaces in parks, and the quality of the environment in residential areas. This study reveals that pandemics directly or indirectly influence the attitudes and behavioral characteristics of people using green park spaces.

The results indicate that changes in UGS visitation behavior can be inferred from participants' demand for UGSs and that visiting UGSs can be effective in improving physiological and psychological health and social interaction development, enhancing public health while promoting harmonious social development. This study highlights the critical and positive role of urban parks during pandemics [46]. However, it is undeniable that the negative effects of the lack of UGSs during quarantine are objective and important. Therefore, it is more important to consider the availability of UGSs, especially those that can be found during special periods. Urban managers can consider building small-scale green spaces in residential areas to mitigate the effects of inaccessibility and lack of access [17] and to encourage people to have more access to outdoor spaces to enhance their physical health [4].

(3) Comparing the type, distribution characteristics, quality, and other characteristics of green spaces in urban public spaces, residents evaluated the green spaces in their area and those in the East Lake Greenway. The increased frequency of visiting UGSs after the pandemic is closely related to the quality of green space distribution in urban public spaces. Additionally, the phenomenon that residents spend less time visiting UGSs shows a positive correlation with the quality of green spaces in parks and a negative correlation with the quality of green spaces in residential areas. In addition, there was a negative correlation between the decline in participants' self-rated mental health status and the quality of green spaces in residential areas. The residents' comparison of green spaces showed that they were generally less satisfied with the green spaces in their area than with the green spaces in the East Lake Greenway Park, and had a higher preference for the East Lake Green Space. The research highlighted public space issues such as uneven distribution of green spaces, differences in green

space types, differences in green space quality, and residents' potential demand for green spaces.

A more in-depth discussion of the issues related to behaviors and attitudes toward urban park use compiled in this study is necessary. Changes in visiting behavior and attitudes after the pandemic were also manifested in the duration and frequency of use of UGSs, travel patterns, and other related aspects. Increased or decreased changes in park use showed the impact of the pandemic on this. In this paper, we start from behavioral perceptions and find that the pandemic affected people's food, clothing, and housing in terms of life and work, but also changed people's habits and ways of using parks. On the one hand, the pandemic changed people's normal life and increased their willingness to visit green spaces. Especially important is that access to UGSs can be effective in improving physical and mental health. On the other hand, the attractiveness of UGSs and the need for their health make access to them inconvenient in this period. In the face of this unexpected situation, we need to: (1) raise awareness and strengthen people's understanding of pandemic and pandemic prevention and control, as well as the benefits of visiting green spaces for physical and mental health; (2) create a good environment, emphasize the positive role of visiting public green spaces, improve the urban green space environment, narrow the gap in the quality of urban public green spaces, highlight the diversity and local characteristics of UGSs, and improve the supporting infrastructure services for urban green spaces; and (3) improve infrastructure services by maintaining or increasing publicly accessible urban green spaces, reassessing our relationship with nature, and resisting future epidemics and pandemics.

## 5. Conclusions

COVID-19 outbreaks and epidemics pose serious challenges to people's productivity and livelihoods. The development of cities, public health, and ecological sustainability are all related to it. Overall, the sudden health crisis is not simple, independent, or minor; it needs to be given sufficient attention and studied. In this study, a quantitative and qualitative approach was used to study 302 participants in Wuhan.

It has been established that the quality of UGSs is crucial to the physical and mental health of residents' lives and the improvement of their well-being. Additionally, our study has made new findings from the association between UGS types and residents' usage patterns and attitudes. The results show that participants' visits to green spaces changed considerably after the COVID-19 outbreak. First of all, the basic information of the participants has changed, basically showing the following characteristics: (1) the gender and age of the participants are concentrated in the group of men and young people; (2) among all the participants, 78.15% are permanent residents, 57.95% own a home, and 68.54% have a bachelor's degree or higher, which allows them to be judged as well educated; (3) in addition to having a stable job, nearly half of the respondents' salaries changed after the pandemic, with 77.27% of those whose salaries changed being lower than before. Second, their perceived health ratings were more significantly related to changes in green space use behavior. Changes in visitors' physical and mental health after the outbreak were significantly correlated with several factors, including the duration and frequency of use of UGSs, travel patterns, and other related aspects.

UGS is not only an important place for regulating the physical and mental health of residents, but also an important ecological resource for the whole city and the world, and plays a regulating role in the ecological crisis of cities as well as the whole human race, and it can be said that the protection and development of UGSs is an epoch-making and historically significant initiative with im-measurable ecological significance. While we draw conclusions from the study, we should also pay attention to the more multifaceted thinking that this issue represents. Therefore, while strengthening the connection between urban residents and UGSs, it is also important to protect UGSs as ecological resources and to help urban parks play a role in sustainable urban development.

Finally, there are some limitations in the methodology, process, and conclusions of this study. This study used questionnaires and on-site research to confirm most of the conjectures, enrich the existing studies, and truly reflect the real usage of residents in Wuhan, the city with the highest concentration of epidemics, under the influence of the pandemic. However, there are still some shortcomings. (1) There are limitations in controlling the whole green space. The UGS we selected is special in that while it represents the park activity characteristics of most city residents, it does not represent the full range of activity characteristics of Wuhan residents. In order to obtain a more comprehensive picture of residents' behavior, more research on other UGSs in different areas is needed. (2) The small sample size and the low participation of the elderly in the study group are also related to the limitations of the location of the selected green spaces. (3) The high educational level of the sample size may be due to the administrative division of the Donghu green space as an educational area in The Wuchang district, with more college students living nearby. However, this study truly reflects the changes in the use of UGSs and provides an effective reference for the next in-depth study.

**Author Contributions:** Funding acquisition, H.W.; methodology, L.L.; project administration, Z.P.; validation, H.W. and Z.S.; writing—original draft, L.C.; writing—review and editing, Z.S. and Z.P. All authors have read and agreed to the published version of the manuscript.

**Funding:** This research was funded by the National Natural Science Foundation of China, grant number 52078390.

**Institutional Review Board Statement:** Ethical review and approval were waived for this study, due to involving no more than minimal risk.

**Informed Consent Statement:** Informed consent was obtained from all subjects involved in the study.

**Data Availability Statement:** Data sharing not applicable.

**Conflicts of Interest:** The authors declare no conflict of interest.

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
