# Peer review of "Change of Residents’ Attitudes and Behaviors toward Urban Green Space Pre- and Post- COVID-19 Pandemic"

_land, doi:10.3390/land11071051_

Round 1
Reviewer 1 Report
This study is an interesting study analyzing changes in users' perceptions related to parks before and after the pandemic.
However, the research question is too simplistic, and it is judged that it is insufficient to properly analyze the situation before and after the Covid-19.
Since this is a paper that has already been surveyed, it is necessary to add a literature review to describe whether the questionnaire used in the current study was properly designed to analyze before and after the Covid-19 situation.
In addition, there is very little comparison with previous studies in the discussion part. Usually, rather than the conclusion part, the discussion part discusses the meaning of the research by citing previous research, and it is common to explain the meaning of the research without citation in the conclusion part.
It is recommended to revise the discussion and conclusion parts as a whole.
Reviewer 2 Report
This manuscript focuses on a critical question regarding the vital role of urban green space pre-and-post Covid-19 pandemic. Overall, the paper has a proper structure and research design. Still, it suffers from several issues that require major revision.
1) The introduction section does not provide sufficient background information on this study. For example, the authors need to introduce the city of Wuhan in the introduction to clarify their research background and motivation.
2) There is a lack of explanation on the location selection. Although the manuscript has a dedicated subsection 2.2.1 to introduce the survey location, it is only a simple introduction of the site without a clear explanation of the reasoning and decision for this site selection. Since the survey is solely based on this site, it is critical to provide convincing evidence to support this decision.
3) This manuscript needs more detailed information on the general population characteristics of the Wuhan or this local district. I find it difficult to draw any conclusion based on the surveyed population characteristics without a clear introduction of the population baseline condition. Therefore, I suggest the authors add more information on the local population's age group and gender distribution.
4) It is not clear how the authors defined the "pre" and "post" pandemic timeline based on the evidence. The manuscript needs to be improved with better descriptions of how they compare "pre" and "post" COVID-19 pandemic in their analysis.
To summarize, this manuscript raises an interesting question on residents' attitudes and behaviors toward urban green spaces before and after Covid-19. However, the current version suffers from a lack of explanation of research design and background information. Therefore, a major revision is needed before being considered for publication.
Reviewer 3 Report
No.1
81-84
”For field survey, the respondents were selected from the core areas of five different scenic spots in East Lake and filled in voluntarily for 6-10 minutes by the respondents in an informed manner. The participants were then asked if they were willing to communicate in depth during the response process, and more detailed interview notes were made. ”
Indicate how many you surveyed.
Also, where are these survey results listed in "3. Results and analysis"? Add a sentence so that we can understand.
No.2
97-106
”The questionnaire was designed to assess three main information: socio-demo-graphic, self-rated of the residents' physical and mental health and the use of UGS. In the first part, socio-demographic information was collected to map the personal situation of the residents in order to connect with the information that follows. In the second part, the self-rated of the residents' physical and mental health was divided into five levels and took the form of individual self-rated, and the results were classified and analyzed. In the third part, six factors, including purpose of visit, frequency of visit, mode of transporta-tion, mode of travel, area visited, and length of stay, which are the most intuitive and reflective of changes in UGS use, were selected for question setting to collect changes in use [34] . ”
It is difficult to understand the specifics of the survey, so please provide detailed information on specific survey items in tables, etc. In particular, you asked UGS before and after COVID-19, so be sure to specify those questions as well. Also, you should add who the target audience is for the online survey. Did you ask users of this park targeted in this study?
No.3
Please describe the results of the "2.2.2. Field Research".
Where in "3. Results and analysis" are these results written? Also, are they different from the interviews in "2.1. Questionnaire survey"? Please add if necessary.
No.4
Research methods are described in "3. Results and analysis".
These methods should also be described in chapter "2. Research methods and data collection".
For example, 182-184
"In this study, questionnaire self-rated was used to obtain the physical and mental 182 health status of the participants. The self-rated was based on Likert Scale Method to clas-183 sify physical and mental health status: (1= poor, 2= fairly poor, 3= average, 4= fair good, 184 5= good), and the results was shown in figure 2. "
No.5
Next, you use a t-test on a Likert scale, is that ok? Of course, You can use it as long as the responses are normally distributed. However, the t-test was used without mentioning this point in this paper. In general, Likert scales are typically used as ordinal scales.
All t-tests should be reanalyzed with another test that can be used for Likert scales.
No.6
"4. Discussion", 311-376.
I cannot find any explanation for Figure 5 in the text. Also, it is difficult to understand what part of the results is relevant to the Discussion.
Please rewrite the Discussion, citing the results obtained to the extent possible and citing other papers.
No.7
"5. Conclusion" 378-419
There are many changes up to line 377. Please rewrite your Conclusion in response to the results. Also, since much is written that is not directly related to your research, please be more explicit about the originality and usefulness of your research.
Round 2
Reviewer 1 Report
The most of previous comments have been corrected. Minor revisions and grammar checks are needed.
Author Response
Thank you for your constructive comments. The English grammar of the manuscript has been checked and revised.
Reviewer 2 Report
The authors have addressed most of the previous comments. There is still one remaining question on site selection that has not been clearly explained. The authors need to provide more explanation and facts to clarify the following questions, either in the Methods section or Discussion section.
- Does the selected site represent urban green spaces (UGS)? My understanding of UGS is a network of green spaces at the city level, including multiple parks, green corridors, and public spaces. Can a single park represent UGS?
- Based on its location, is the selected park qualified as an "urban" green space? The map indicates that it is located on the outskirt of the central urban area, so why not select green spaces with more urban characteristics?
- Do people's attitudes towards this park generally represent their attitude to UGS?
Besides the above questions, this manuscript needs minor revisions and grammar checks to improve the English writing.
Reviewer 3 Report
 I believe your study's results will be useful in planning the subject area.
Author Response
Thanks for your encouraging comments on the merits of this manuscript.